# Epilepsy Networks and Their Surgical Relevance

**DOI:** 10.3390/brainsci14010031

**Published:** 2023-12-28

**Authors:** Kevin Hines, Chengyuan Wu

**Affiliations:** Department of Neurosurgery, Thomas Jefferson University Hospital, Philadelphia, PA 19107, USA; chengyuan.wu@jefferson.edu

**Keywords:** epilepsy, network, resection, ablation, neuromodulation, invasive monitoring, sEEG

## Abstract

Surgical epilepsy is a rapidly evolved field. As the understanding and concepts of epilepsy shift towards a network disorder, surgical outcomes may shed light on numerous components of these systems. This review documents the evolution of the understanding of epilepsy networks and examines the data generated by resective, ablative, neuromodulation, and invasive monitoring surgeries in epilepsy patients. As these network tools are better integrated into epilepsy practice, they may eventually inform surgical decisions and improve clinical outcomes.

## 1. Introduction

This paper is a narrative review of brain networks in epilepsy and their recent implications in epilepsy surgery. To understand the trend towards understanding epilepsy as a “network disease”, clinicians must understand network epilepsy as multiple regions in the brain involved in the development and propagation of seizures [1,2,3]. These multiple brain regions may be connected both structurally and functionally with varying temporal patterns of spread between regions. In addition, these areas are also connected to a normally functioning brain. It is through the definition of the epileptogenic tissue, as well as the connections that allow seizures to propagate elsewhere, that an epileptogenic network may be defined. As diagnostic and invasive studies have improved, clinicians have developed more insight into these mechanisms. These studies include examples such as tractography (Figure 1), functional magnetic resonsance imaging (MRI) (Figure 2), and invasive interrogations like stereo electroencephalography (Figure 3). Tractography delineates structural connectivity via MRI diffusion sequences. Functional MRI allows for a large-scale sampling of physiological brain coupling via the visualization of multiple brain regions and their associated activity at different times. Stereo electroencephalography provides a better temporal resolution of neurophysiological signal spreading from specific point to point based on the surgical implantation of specific brain regions. All studies contribute to patient specific insights regarding epileptogenic regions of the brain and how they connect to each other and allow the spread of seizures. A better understanding of these networks allows clinicians to better plan how to disrupt them and improve surgical outcomes in epilepsy patients.

## 2. Evolution of Network Epilepsy

Initially thought to be of religious or magical consequence, the understanding of epilepsy has drastically improved over time. With descriptions dating back as far as 1050 BC in Babylonian text, seizures and epilepsy have been described and explored by many [4]. It was not until neurology and epileptology were established in the 19th century that epilepsy was more systematically characterized. Scientists such as John Hughlings Jackson, Robert Todd, Louis Francois Bravais, and Theodore Herpin established seizure types as well as a categorization of the seizures based on which “level” of the brain the seizure originated [4]. Building on this, Paul Broca combined his anthropological background with his interest in infantile seizures to propose trephination as a treatment for epilepsy [5]. As understanding grew, it became clear that focal brain pathology may result in epilepsy. As such, semiology and neurological deficits associated with such pathology began informing epilepsy surgeons when developing laterality and general localization hypotheses [6].

However, the 20th century saw the development of technology that enabled a more accurate localization and characterization of epilepsies. Neuroscientist Herbert H. Jasper pioneered the use of the electroencephalograph (EEG) as a tool to localize epilepsy prior to surgical intervention. It was through his efforts with Wilder Penfield that patients’ epilepsies were preoperatively localized so that craniotomies could be performed. In many cases, lesional pathology was resected, alleviating the epilepsy [7]. Through the use of large craniotomies for surface mapping with electrocorticography (invasive EEG), Jasper and Penfield saw that the removal of tissue to attenuate abnormal spikes prognosticated good seizure outcomes in many of their patients. Rasmussen expanded on this with his localization concepts: primary localization was achieved by invasive EEG for the mapping of resection; secondary localization involved understanding cortical recruitment and spread during seizures; and tertiary localization defined the extent of resection necessary to produce satisfactory seizure reduction or cure [8].

While early results were promising, it was noted that EEG methods were not sufficient to localize and resect epileptogenic areas of the cortex while still producing favorable seizure outcomes in a number of cases. As such, the 1960s gave rise to Talaraich and Bancaud’s philosophy that clinical, anatomical, and electrophysiologic correlation all together took precedence over surgery solely informed by EEG [9]. This approach facilitated the existence of the epileptogenic zone, which reflects the cortical site of potential initiation and the organization of activity into seizures. This philosophy gave rise to the advent of stereo-electroencephalography, allowing for a three-dimensional interrogation of the spatiotemporal organization of seizures.

In the 1990s, Hans Luders and colleagues modified the term epileptogenic zone to encompass “the area of cortex that is necessary and sufficient for initiating seizures and whose removal (or disconnection) is necessary for complete abolition of seizures” [10]. This definition shifted towards a more conceptual understanding of the goals of surgery to account for the difference between the epileptogenic zone and ictal onset zone. Given the continued failures in epilepsy surgery related to the focal control of the disease process, Susan Spencer advocated for efforts to affect larger networks [11]. The challenge there remains to identify and separate pathologic epileptogenic tissue from normal but affected networks modulated by the epileptogenic zone. As such, some groups have attempted to quantify the recorded anatomy’s contribution to the epilepsy network. In one such example, Bartolomei et al. coined the term epileptogenicity index (EI) in their exploration of temporal lobe epilepsies [12]. Using the spectral and temporal characteristics of the structures interrogated on EEG, the contribution to the epilepsy network was quantified. Similarly, in response to the work by Spencer and Bartolomei, John Miller has also advocated for a replacement of the EZ with the “seizure generating network” to explain the dynamic clinical expression of seizures, resection failures, early propagation, and success of newer approaches [13].

As this approach to epilepsy has gained validity, recent investigations have primed the field for a discussion of both epileptogenic tissue and the involved epilepsy network rather than anatomic regions alone. These network relationships inform surgical treatment. One example includes temporal lobe epilepsy. While seizure freedom rates after anterior temporal lobectomy are accepted as high [14,15], surgical failures prompt discussion as to better preoperative diagnosis or surgical planning. This has led to the evaluation of “temporal plus epilepsy”. Rather than the temporal lobe anatomically acting as the epileptogenic zone, its network connections to other circuits prevents cure after the resection of the temporal lobe and mesial structures alone [16]. The potential integration of the insular [17], orbitofrontal [18], and occipital networks [19] raises the question of more extensive resections/disconnections to prevent surgical failures in temporal lobe epilepsy patients.

Similar to the well-studied temporal lobe epilepsy, epileptogenic zones arising elsewhere anatomically have the potential for larger network involvement and need for consideration in surgical planning. Occipital epilepsies have the well-documented spread and network involvement of temporal or parieto-frontal networks [20,21,22]. Insular epilepsies are known to involve the opercular cortex as well as the supplementary motor area cortex [23,24]. The cingulate gyrus network participates in the limbic circuit and even parieto-occipital regions in epilepsy patients [25,26]. These are just several examples of the emerging network relationships that exist in epilepsy patients that may affect surgical prognostication and outcomes. There are many more networks under investigation that need to be accounted for in the current paradigm of epilepsy surgery. These concepts have paved the way for neuromodulation in epilepsy as well as more network-based investigations of networks and surgical treatments. Given the shift towards network epilepsy, we aim to review recent advances in knowledge of the epileptogenic networks derived from epilepsy surgery.

## 3. Networks in Resective and Ablative Epilepsy Surgery

In the populations of surgical patients who have undergone advanced imaging such as functional MRI or tractography, the response to surgery may be correlated with connectivity to discern the areas participating in an epileptogenic network. In 2019, Alizadeh et al. examined tractography density differences between pathologic and healthy hemispheres in patients undergoing LITT or ATL for temporal lobe epilepsy (TLE) [27]. They found that epilepsy patients with increased white matter density in the ipsilateral lingual, temporal pole, pars opercularis, inferior parietal, and contralateral frontal pole segments were more likely to have residual seizures after surgery. They also noted that patients with left versus right mesial temporal sclerosis (MTS) exhibited differences in connectivity patterns. Bilateral and widespread white matter changes were also more prominent in left MTS compared to right-side MTS. This provides insight that even the laterality of seizure onset may affect network involvement. Network information in the form of resting state fMRI and EEG data was leveraged by Neal et al. in 2020 to map a hypothetical irritative zone in patients undergoing temporal lobectomy [28]. The group compared the preoperative network connectivity to that of the irritative zone in seizure-free and non-seizure-free patients. They also compared temporal lobe connectivity in patients undergoing temporal lobectomy to healthy controls. They found that temporal lobe networks in epilepsy patients demonstrated higher connectivity than healthy controls. When comparing the network connectivity in seizure-free vs. non-seizure-free outcomes, the authors noted that a higher percentage disruption of epilepsy networks was more closely associated with seizure-free outcomes. Finally, patients with a greater disruption of epileptogenic networks demonstrated improvements in quantitative neuropsychological testing. Liang et al. also surgically demonstrated that disconnection, as opposed to resection, affects epilepsy networks [29]. Functional connectivity in 30 Lennox Gastaut Syndrome patients was derived from EEG data before and after callosotomy. Patients with good seizure outcomes had a shifting of the “hubs” of connectivity on the EEG from paramedian regions to a more lateral cortex. Patients with poor seizure outcomes did not see this shift towards a more homogenously connected state as the hubs remained all paramedian. Together, these studies suggest the presence of increased connectivity in abnormal epileptic networks and the surgical disruption of those networks correlated with surgical outcomes. Further bolstering the need for network-based approaches to resection are the results of Josephson et al.’s systematic review of the outcomes in mesial temporal lobe epilepsy (mTLE). Despite the focus on mTLE, the authors found that standard resection including neocortical removal conferred higher seizure freedom rates than the selective resection methods such as transylvian or transcortical amygdalohippocampectomies [30]. This is presumed to be related to the disruption of a wider set of network pathways that seizures may originate in or spread through.

In addition to connectivity studies, the examination of resected anatomy associated with seizure freedom provides information on structural involvement in network epilepsy. Recent advances in the understanding of the piriform cortex have supported this notion. In 2019, Galovic et al. found that in 107 patients undergoing temporal lobectomy for unilateral TLE, the proportion of the piriform cortex resected was directly associated with seizure freedom [31]. As no other structure shared this association between the extent of resection and the outcome, this finding alludes to the role that the piriform cortex may play in the generation and propagation of seizures in temporal lobe network epilepsy. Until recently, the piriform cortex has not received much attention in human epileptology. It has inputs from the olfactory bulb, other olfactory cortical areas, and the contralateral piriform cortex. In addition, its outputs include strong limbic connections such as the amygdala and entorhinal cortex, orbitofrontal cortex, insula, and even mediodorsal nucleus of the thalamus as well as the hypothalamus [32]. This demonstrates that even small structures may play key network roles in epileptogenesis if highly connected. Similar to temporal lobe epilepsy, such concepts may be applied to other regions. Giampiccolo et al. used tractography in 47 patients undergoing frontal lobectomy to identify the “disconnectome” and predict late seizure recurrence [33]. They found that long-term seizure freedom as durable as five years was associated with the disconnection of anterior thalamic and cortico-striatal fibers. In addition to the epileptogenic frontal lobe resection, the disconnection of the epileptogenic network was crucial in the prevention of recurrence.

The utilization of minimally-invasive ablative surgery in epilepsy has also led to a greater understanding of epilepsy networks. As opposed to resection, the ablation of a seizure focus lends itself to more accurate postoperative volumetric analysis because the ablation and remaining tissue are less prone to brain shift or postoperative distortion. As such, volumetric analysis after laser interstitial thermal therapy (LITT) has yielded a depth of information on networks in epilepsy. For example, by normalizing ablation cavities in 175 patients undergoing LITT for mTLE to a common atlas space, Wu et al. demonstrated that the ablation of anterior, inferior, and mesial structures is more associated with seizure freedom while ablation posterior the mesencephalic sulcus provides diminishing returns with respect to seizure freedom [34]. When examining the structures associated with seizure freedom, this analysis also supports the recent development in the epilepsy literature that the piriform cortex is strongly associated with epileptogenic networks and seizure outcomes after epilepsy surgery for mTLE. As we continue to utilize volumetric analysis for ablative or resective lesions in epilepsy, we will improve our understanding of which structures are associated with seizure freedom and thus are implicated in network epilepsy.

## 4. Networks in Neuromodulation Epilepsy Surgery

While resection or ablation may provide information by exclusion about what structures were previously participating in epilepsy networks, neuromodulation procedures provide information about structures actively included in the network by modulating their function. One of the oldest forms of neuromodulation in the treatment of epilepsy is vagal nerve stimulation (VNS). Investigators found that the cycled stimulation of the vagal nerve decreases frequency by at least 50% in 30–50% of patients with an increasing effect over time [35,36,37]. While the mechanisms of seizure reduction are still under investigation, VNS is thought to lower seizure burden by brainstem, limbic, and cortical modulation through the vagal afferent network. Cortically, VNS alters connectivity and this alteration in network connectivity contributes to the increasing and sustained efficacy over time [38]. In patients who have surgically implanted VNS, network alteration has been documented post-operatively. Zhu et al. found that when compared to their preoperative altered regional activity on resting state fMRI, patients implanted with VNS had reorganization and increased regional homogeneity in superior and middle temporal gyri that correlated with an improved seizure outcome after 3 months of stimulation [39]. Similarly, Wang et al. found that after 6 months of stimulation, VNS patients reorganized resting state networks into more regionally homogenous states and demonstrated a suppression of excessive salience network activation [40]. As experience with neuromodulation and VNS grows, a better understanding of network remodeling will lead to better patient selection and prediction of seizure outcomes in response to VNS therapy.

An increasingly common example of the neuromodulation approach is deep brain stimulation (DBS) for treatment of epilepsy. When epileptogenic tissue is too diffuse, in eloquent tissue, or otherwise unable to be resected, the modulation of the network may be the patient’s best opportunity for the palliation of epilepsy. The most widely used target for DBS in epilepsy is the anterior nucleus of the thalamus (ANT). The stimulation of ANT has been shown to modulate frontal and temporal circuits on EEG [41]. By targeting the ANT, the limbic circuitry including the mammillothalamic tract, cingulate, hippocampus, and parahippocampus may be modulated. In the SANTE trial in 2010, it was demonstrated that patients have a 41% seizure reduction within 1 year of implantation [42]. After 5 years of treatment and programming optimization, the median seizure reduction increased to 69% [43]. An interrogation of the functional network via EEG after DBS has provided some biomarkers for DBS treatment in epilepsy. Scherer et al. found that responders to DBS were more likely to demonstrate widespread cortical desynchronization of alpha and theta bands whereas non-responders did not [44]. In addition to cortical changes, newer sensing technology allows for the interrogation of thalamic local field potentials in epilepsy patients who have undergone DBS implantation. While the potential for clinical integration remains high as this may allow for closed loop DBS in the future, consistent biomarkers remain elusive as no single frequency bands appear to differentiate ictal from non-ictal activity [45]. While large studies investigating the stimulation effect on epilepsy networks are limited, several centers have begun to investigate functional MRI differences between on and off stimulation in the ANT [46,47,48]. This work has demonstrated that stimulation increases in the activity of limbic structures such as the bilateral thalamus, anterior and posterior cingulate cortex, amygdala, and hippocampus. In addition, the default mode network areas such as the precuneus and medial prefrontal cortex showed increases in activity during stimulation. Other research has demonstrated network modulation when examining the proximity of an estimated field of electrical stimulation to the mammillothalamic tract (MTT). Specifically, Schaper et al. noted that responders to ANT DBS were more likely to have stimulation closer to the MTT [49]. Given the activity seen in limbic circuitry including Papez’s traditional components, it is not surprising that the MTT white matter tracts may play a large role in temporal lobe epilepsies and their treatment. These data lay the foundation for patient-specific epilepsy networks that can be used to inform response to therapy. These methodologies may be adapted to larger studies investigating long-term network effects of deep brain stimulation. At this time, there is unfortunately a dearth of literature in patients undergoing alternative thalamic target stimulation (such as centromedian or pulvinar nuclei). As more patients undergo DBS for centromedian [50,51] or pulvinar stimulation [52], on/off testing in combination with EEG and radiographic data may demonstrate variable network modulation and thus differentiate target utility in a patient-specific manner. While stimulation-related functional data are being generated, structural investigations elucidating the connectivity of these targets is already underway. Using a finite element model approach, Diaz et al. estimated the volume of tissue activation (VTA) in 10 patients who underwent DBS CM placement for treatment of generalized pharmacoresistant epilepsy [50]. This VTA was used to seed both tractography and functional MRI studies. They demonstrated that responders were correlated with VTA-modulating reconstructed networks that included a sensorimotor, supplementary motor area, cerebellum/brainstem, and reticular activating network. Similarly, as the pulvinar nucleus is being investigated as a target, its network implications must also be investigated. Leh et al. used diffusion tensor imaging (DTI) in six healthy controls to construct pulvinar fiber tracts and found that the pulvinar nucleus is interconnected with the thalamus, caudate, as well as cortical areas including visual, associative visual, posterior parietal, inferior visual temporal, and frontal eye field cortical areas [53]. Large as it is, it is important to note that the pulvinar subregions have different connectivity as more lateral regions are more closely connected with posterior quadrant networks [54]. As potential targets for neuromodulation in epilepsy arise, it is important to interrogate them via functional and structural testing for a more complete understanding of their therapeutic mechanisms.

While the nodes of thalamic connectivity move towards informing DBS targeting, responsive neurostimulation (RNS) employs closed loop stimulation to modulate epilepsy networks [55]. Similar to other forms of neuromodulation, the efficacy increases over time as neuromodulation alters epilepsy networks [56]. To further support the network modulation effects of RNS, Khambati et al. retrospectively analyzed interictal ambulator intracranial EEG readings at each contact [57]. By categorizing which contacts were in the seizure network, the authors used the theta, alpha, beta, and gamma bands to estimate functional connectivity. Beta bands were noted to diverge within network foci in responders, with the earliest evidence of increased beta connectivity appearing around 270 days in responders. In addition, theta, alpha, and gamma bands diverged between foci. This study provides evidence in the changes in functional connectivity occurring after sustained responsive neurostimulator treatment. As experience with RNS has accumulated, investigators note that patient-specific structural and functional connectivity may correlate with seizure reduction and epilepsy outcomes. Charlebois et al. noted that patient-specific tractography and connectivity was correlated with RNS effectiveness. The increased connectivity of the estimated stimulation field to the medial prefrontal cortex, cingulate cortex, and precuneus was associated with larger seizure reductions [58]. The highly connected pulvinar nucleus has already been employed in a closed-loop manner with success. Using it to modulate the network connections as described above, Burdette et al. described the placement of pulvinar depth electrodes paired with occipitoparietal leads in an RNS system to treat three cases of posterior quadrant epilepsy [59]. All patients responded during follow-up and two had 90% or greater reduction in seizure frequency. While these results require rigorous verification, these promising results support using network information in RNS and epilepsy surgery.

## 5. Networks in Stereoelectroencephalography

In addition to destructive and neuromodulatory surgical insights, invasive monitoring also has yielded significant surgical information on network epilepsy. Stereoelectroencephalography (sEEG) allows for the physiological examination of functional connectivity with excellent temporal resolution compared to other modalities such as rs-fMRI [60]. Azeem et al. combined sEEG with probabilistic white matter tractography. By seeding regions of interest with spike propagation relationships, they were able to compare white matter connections between the two points in epilepsy patients and matched controls from the Human Connectome Project [61]. They demonstrated that spike propagation in epilepsy patients was associated with a higher likelihood of direct white matter connection. As tractography and invasive monitoring methods improve, more indirect inferences may be drawn from combining the two modalities. Mitsuhashi et al. generated a novel dynamic tractography model to localize interictal spike sources and monosynaptic spike propagation through the white matter in sEEG [62]. In their model, they found that spike sources were more likely to be included in the resection in patients who achieved ILAE class 1 outcomes. This manner of combining structural tractography data and physiological sEEG data may aid in surgical decision making after rigorous validation by surgical results. Koubeissi et al. used invasive monitoring in combination with forniceal-hippocampal-evoked responses to examine functional connectivity as well as the efficacy of forniceal stimulation for the reduction of seizures [63]. They noted that hippocampal-evoked responses as well as posterior cingulate-evoked responses during forniceal low frequency stimulation demonstrated the intimate connectivity of the memory circuit and default mode network. This effect decreased seizure frequency after low frequency stimulation without compromising memory function. Not only did these epilepsy networks correlate closely with memory and default mode networks, but this study also demonstrated differential network effects based on the type of stimulation applied. Such effects of differential stimulation on networks will be paramount in defining these networks during invasive monitoring.

In addition to combining sEEG with advanced imaging modalities or stimulation, sEEG mediated radiofrequency ablation has been used to disrupt epilepsy networks. This technique allows neurologists and neurosurgeons to review intracranial EEG findings after implantation and while the leads are still in place, use radiofrequency ablation to target epileptogenic contacts noted in recordings [40]. This is beneficial because the intervention is based on current epilepsy network data without the need to insert a probe or move the contacts prior to direct intervention. In addition, the ablation is performed with the patient awake so that as the current is applied to each target contact, the surgeons may monitor for adverse effects. While long-term seizure freedom rates are generally reported at low rates [64,65,66], the risk and additional burden to the patient that RF ablation imposes is minimal. It is particularly effective in some disease processes that may have been considered inoperable in the past. In patients with periventricular heterotopia (PVH), sEEG allows for the interrogation of multiple lesions that could be epileptogenic, many of which are poorly accessible. A review of sEEG-mediated RF ablation in PVH shows reasonable response rates at 81% and even a 38% seizure freedom rate [65]. Lastly, even if patients who have sustained seizure freedom for several months recur, the effect of ablation has a high positive predictive value for the success of subsequent lesional epilepsy surgery. It is a powerful tool to validate a surgical hypothesis and proposed resection [67].

In addition to monitoring the white matter tracts involved in epilepsy networks, there has been increased interest in monitoring deeper nodal structures such as the thalamus. Illyas et al. utilized extensions of sEEG electrode trajectories to include thalamic nuclei for monitoring [68]. Their targets included the anterior (ANT), centromedian (CM), and mediodorsal (MD) nuclei of the thalamus. During seizure activity, they observed changes in high frequency activity (HFA) based on the target nucleus and various ictal time points. While ANT and MD were noted to have greater HFA at seizure onset, CM HFA activity peaked towards the end of seizure activity. These findings suggest that the CM may play a role in seizure termination. In a similar study, Soulier et al. monitored the ANT and pulvinar nucleus (PUL) in TLE patients undergoing sEEG. They noted that the nodal “in” connectivity, as defined by the EEG propagation of the signal towards the target nucleus, of both increased as the seizure progressed after onset inferring that these thalamic nuclei may act as more of a sensor than propagator in focal TLE. Interestingly, they did also note that the PUL outward connectivity increased during synchronous activity at the end of recorded seizures. In this instance, PUL may play a key network role in seizure termination versus propagation [69].

Lastly, sEEG of thalamic monitoring may allow for more rigorous phenotyping of seizures and their associated networks. Wu et al. targeted the thalamic nuclei during sEEG implantation of multiple epilepsy types to simultaneously interrogate different SOZs as well as thalamic PUL, ANT, and MD nuclei [70]. By categorization of seizure by spread types and onset location, 20 of the 22 seizures were noted to involve thalamic nuclei. The only exception was one seizure that remained focal and another that spread to the contralateral cortex without any recorded thalamic involvement. Even more interestingly, thalamic nuclei EEG morphologies were similar among the same seizure types. The similar thalamic EEG signatures among seizure types allows for a future investigation of elucidating seizure networks by thalamic involvement or even the prediction of an ideal neuromodulation target after sEEG.

## 6. Conclusions

The surgical treatment of epilepsy continues to evolve as new treatment paradigms are developed. Through the use of a network approach, understanding of the pathophysiology and surgical outcomes continues to improve. The resection or ablation of highly involved nodal networks reduces seizure recurrence. The neuromodulation of various targets reduces seizure frequency while allowing the study of involved cerebral circuitry and white matter pathways. Finally, invasive diagnostics such as sEEG also provide valuable information on epilepsy networks that may inform future surgical decision making. As research is conducted to standardize radiographic technique and larger databases are accumulated, it is likely that surgeons will be able to leverage this information to improve seizure outcomes in their patients. The future of epilepsy surgery will be not only defining surgical targets as an anatomical epileptogenic zone, but also delineating involved networks to ensure an adequate disruption of seizure spread through highly connected neurons.

## Figures and Tables

**Figure 1 brainsci-14-00031-f001:**
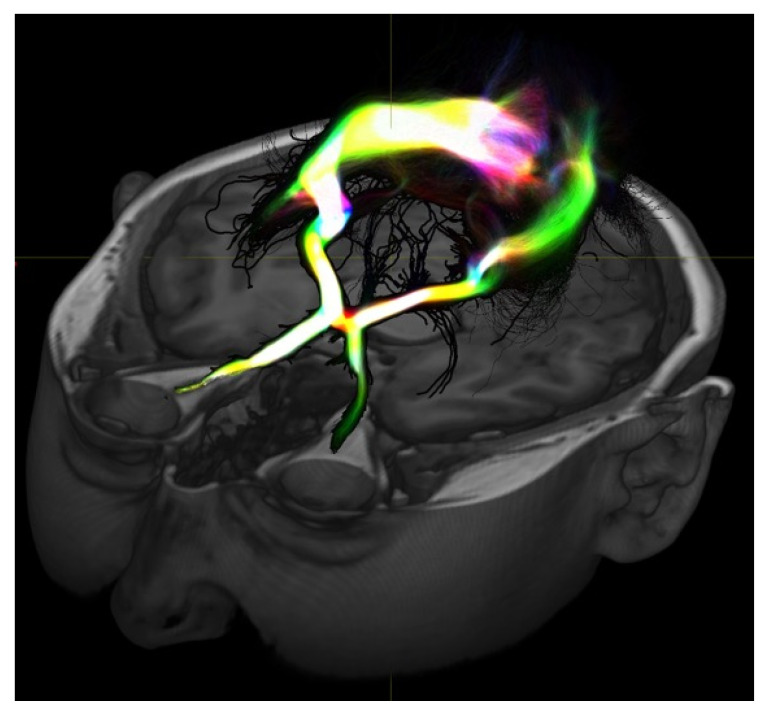
Diffusion-based MRI imaging allows for visualization of white matter structure and here shows the visual pathway. These types of studies may provide information on structural connectivity between areas of interest to help define connections or epileptogenic tissue or even normal tissue to preserve during resection.

**Figure 2 brainsci-14-00031-f002:**
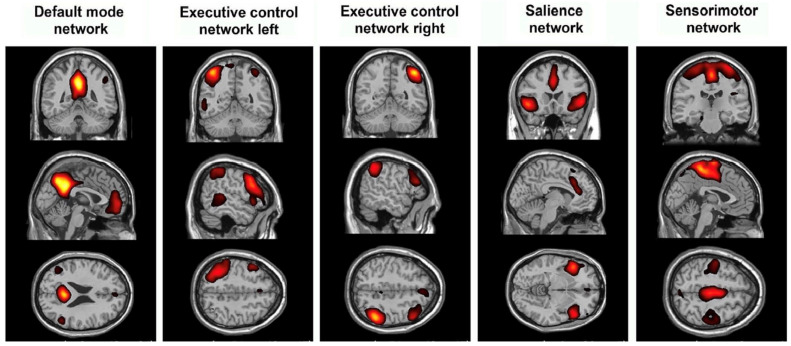
Differences in resting state functional MRI may show how epilepsy modulates established networks and which networks are involved in seizure propagation or generation.

**Figure 3 brainsci-14-00031-f003:**
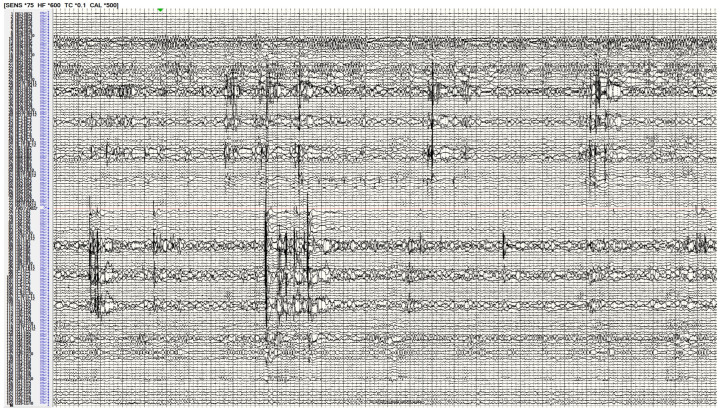
Stereo-electroencephalography interrogation of a patient with bilateral temporal lobe epilepsy. The recording demonstrates bilateral hippocampal spikes implying an epilepsy network involving both these regions.

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
