# Peer review of "Epilepsy Networks and Their Surgical Relevance"

_brainsci, 2023, doi:10.3390/brainsci14010031_

Round 1

Reviewer 1 Report

Comments and Suggestions for Authors

The topic of the manuscript is extremely relevant, due to the fact that epilepsy surgery is constantly evolving and is sometimes the only effective method for treating drug-resistant epilepsy. During the review process of the manuscript, a number of comments arose:

1) Considering that the manuscript is devoted to information about epileptic networks, the authors should devote a separate section with an appropriate subtitle to the description of epileptic networks. In which it is necessary to disclose the following information: definition of epileptic networks, examples of localization and prevalence of epileptic networks with different localizations of epileptic structural and functional brain damage, etc.

2) Also in this section, for greater clarity, it is necessary to provide diagrams or pictures of various prevalence and localization of epileptic networks.

Author Response

The authors thank Reviewer 1 for taking the time to evaluate the manuscript and the valuable feedback provided.

  1. We totally agree with Reviewer 1's feedback regarding defining epilepsy network and have done so at the beginning of the paper to provide a foundation for readers going forward in the manuscript.
  2. Some figures to illustrate examples of "network information" have been provided so readers may understand how some of this information is generated and applied.

Reviewer 2 Report

Comments and Suggestions for Authors

The review on Network Information in Epilepsy Surgery makes a serious effort to gather an overview of how reserach within the field of epilepsy surgery has contributed to the understanding of plasticity and function of brain connectivity in seizure generation and spread. The review also summarizes how epilepsy surgery can benefit from information on network function. 

The review is timely and adresses very important issues within the field.

However, methods used to find articles cited in the present manuscript is not described and this review can not strictly be classified as a systematic review. No meta analysis is performed.  Although the reference list is neither complete nor very extensive, major findings are covered.

The topic is complex and, although the content and facts presented are adequate, the review would benefit from a clear and revised structure. Explanations and reasoning can be more elaborate, simplifying reading. 

There is a complete lack of illustrations, tables and figures. The review can be much improved by the use of tables and figures.

A list of abbreviations is warranted.

Comments on the Quality of English Language

Language is surprinsingly poor as I presume that the first author is a native English speaker. The language clearly needs editing. Some of the errors makes the text somewhat hard to understand. Sentences can be shortened and made clearer.

Author Response

The authors thank Reviewer 2 for taking the time to provide feedback on the manuscript.

  1. The manuscript was an invited narrative review to a special issue of Brain Sciences thus the goal was not to perform a meta-analysis. However the authors agree this should be clarified in the manuscript and has been added at the beginning of the paper.
  2. Editing/revision of sentence structure has been performed to increase clarity. In addition, network epilepsy has been defined in the introduction of the paper to provide greater understanding going forward.
  3. The authors agree figures would improve the paper's quality. Figures showing examples of epilepsy network demonstrations have been included.
  4. A list of abbreviations has been submitted.

Reviewer 3 Report

Comments and Suggestions for Authors

I consider that the topic of the document is definitely, interesting. Neverthless, in my concern, authors speak about epileptic networks however, such "networks" are never offered nor described. I had consider that authors were  speaking about mathematical entities when they use the term "epileptic networks". See for instance page 3, lines 116-123:

-----------

"...outcomes. Finally, patients with greater disruption of epileptogenic net-116 works demonstrated improvements in quantitative neuropsychological testing. Liang et

117 al also surgically demonstrated that disconnection, as opposed to resection, also affects

118 epilepsy network26. Functional connectivity in 30 Lennox Gastaut Syndrome patients

119 was derived from EEG data before and after callosotomy. Patients with good seizure

120 outcomes had shifting of “hubs” of connectivity on EEG from paramedian regions to

121 more lateral cortex. Patients with poor seizure outcomes did not see this shift towards a

122 more homogenously connected state as the hubs remained all

---------

From my perspective, the description is too qualitative without offering any objetcive way to conceive/evalaute what authors describe here (and this is just an example). What I mean is that, from the title of the submission and from the narrative, I (and I 'd say any reader) would imagine a mathematical description or model or structure that could reflect what is narrated here. In my concern. although the document describes "networks" there is no objective way to conceive them, to evaluate their behavior, (with/without antiepileptic drugs for instance). At least, if mathematics/engineering is not the field of authors. they should provide figures, diagrams or something that actually, really shows what a network is. Even description of math/elec. eng. models could be cited but all this chronicle (with all respect) resembles a fairy tale because (reiteration) all the description lies in lines and lines of descriptions. Actually, the scope of the journal supports me: 

#########

"Our aim is to encourage scientists to publish their experimental and theoretical results in as much detail as is required to fully convey the information. There is no restriction on the maximum length of the papers. Electronic files reporting the extended details of computational, statistical, or experimental procedures, if unwieldy for presentation in the body of the manuscript, can be deposited as supplementary material."

#########

A random review of some papers (already published) permits to notice that, in spite of peculiar narrative characteristics of them, numerical data, figures and diagrams support the core of the document. I publish in the field of epilpetic disorders from a numerical and signal processing perspective and I understand that this is not the field of some people but anyway, an author have to properly support what they show.

Finally, I advise to support this document with tables, numbers. diagrams, EEGs,  etc to avoid a vacuous description of a brain network, particularly in epilepsy. A simple narrative is not enough. Books and papers abt it provide that. Imagine a Doose or Lennox-Gastaut description without EEGs, diagrams, photos, etc. Ypu can show brain images and networks on it. 

Author Response

The authors appreciate Reviewer 3's time and feedback in evaluating the manuscript.

  1. The authors agree defining network epilepsy would improve the quality of the manuscript. A definition of network epilepsy has been added in the beginning of the paper to frame the following discussion.
  2. The authors have submitted an invited narrative review to a special issue of Brain Sciences. To clarify the objective of the paper for readers, the title has been altered. Figures demonstrating the network information pertinent to clinical decision making have been added (ie sEEG output).
  3. Figures (in addition to the altered introduction) have been added to avoid a vacuous description of epilepsy networks.

Round 2

Reviewer 3 Report

Comments and Suggestions for Authors

I uploaded a list of remarks. Please add to the draft that list showing in the corrected document what you did to improve it according to those remarks. 

Please notice that it is very difficult to follow what you did in the PDF presented in this way. 

Author Response

The authors agree that highlighted changes would certainly be helpful! We had an email exchange with a managing editor where a copy was sent and was told it would be uploaded and available for reviewers. We will now attach it here as well for you review!